# Evaluation of Antimicrobial and Antibiofilm Activity of *Eucalyptus urograndis* (Clone I144) Pyroligneous Extract on Bovine Mastitis Isolate of Multiple-Drug-Resistant *Staphylococcus aureus* Strains

**DOI:** 10.3390/microorganisms13122771

**Published:** 2025-12-05

**Authors:** Isadora Karoline de Melo, Caio Sergio Santos, Nilza Dutra Alves, Gustavo Lopes Araujo, Aline Maciel Clarindo, Alexandre Santos Pimenta, Denny Parente de Sá Barreto Maia Leite, Rinaldo Aparecido Mota, Francisco Marlon Carneiro Feijó

**Affiliations:** 1Center of Agrarian Sciences, Federal Rural University of the Semi-Arid—UFERSA, Av. Francisco Mota, 572, Bairro Costa e Silva, Mossoró 59625-900, RN, Brazil; isadorakmelo@gmail.com (I.K.d.M.); caiosergio@ufersa.edu.br (C.S.S.); gustavo.araujo51015@alunos.ufersa.edu.br (G.L.A.); aline.clarindo@alunos.ufersa.edu.br (A.M.C.); 2Graduate Program in Forest Sciences—PPGCFL, Federal University of Rio Grande do Norte (UFRN), Rodovia RN 160, km 03 s/n, Distrito de Jundiaí, Macaíba 59280-000, RN, Brazil; alexandre.pimenta@ufrn.br; 3Department of Veterinary Medicine, University Federal Rural of Pernambuco, Recife 52171-900, PE, Brazil; dennyparente@hotmail.com (D.P.d.S.B.M.L.); rinaldomota9@gmail.com (R.A.M.)

**Keywords:** antimicrobial agent, antibiotic, bacterial resistance, herbal medicines

## Abstract

Milk is an important agricultural product and is consumed worldwide. However, the dairy sector faces a significant challenge due to bovine mastitis, a common disease that has a substantial impact on the dairy industry. In more severe cases, it leads to the culling of chronically infected cows. Mastitis poses a risk due to the frequent use of antibiotics in treatment, which contributes to the spread of bacteria with antimicrobial resistance. The present study aimed to evaluate the antimicrobial and antibiofilm potential of a pyroligneous extract of *Eucalyptus urograndis* (clone I144) against multidrug-resistant *Staphylococcus aureus*, the causative agent of mastitis. Sensitivity profiles to various conventional antibiotics were assessed, including the minimum inhibitory concentration (MIC), the minimum bactericidal concentration (MBC), and biofilm inhibition, in ten *Staphylococcus aureus* strains using the crystal violet method. The results showed that the multidrug-resistant strains were sensitive to the pyroligneous extract of *Eucalyptus urograndis* (clone I144) at a concentration of 12.5% and exhibited antibiofilm activity starting at a concentration of 3.13%. In conclusion, our findings show that the pyroligneous extract of *Eucalyptus urograndis* (clone I144), at 12.5%, inhibited different multidrug-resistant *S. aureus* and MRSA strains isolated from bovine mastitis. These results indicate that the extract represents an effective preventive strategy against mastitis-causing pathogens that are difficult to treat, making it a promising alternative to reduce the dependence on synthetic antibiotics. In vivo studies are needed to confirm these findings and provide a basis for evidence-based clinical guidelines.

## 1. Introduction

Milk is an important agricultural product and is consumed worldwide. The global production of bovine milk has increased to approximately 950 million metric tons in recent years [1]. However, the dairy sector faces a significant challenge due to bovine mastitis. This disease occurs frequently and significantly affects the dairy and economic sectors, resulting in financial losses with the use of medicines, reduced milk production, and changes in milk composition, which can lead to the disposal of the product and, in more severe cases, the slaughter of chronically infected cows [2,3].

Bovine mastitis is an inflammation of the mammary gland caused by bacterial invasion, representing one of the most urgent challenges for dairy farming worldwide [4]. Its impacts go beyond significant economic losses in milk production, also affecting public health, mainly due to the frequent use of antibiotics in the treatment of the disease. Mastitis can manifest in a clinical form, with visible signs in the milk and udder, or in a subclinical, asymptomatic form, which is equally harmful to production [5,6,7].

Therefore, the effective control of mastitis is crucial to maintaining animal health and, consequently, protecting consumers of dairy products. Interdisciplinary studies and integrated management measures are necessary to mitigate the risks associated with mastitis, fostering sustainable and safe dairy production practices.

*Staphylococcus aureus* is considered the primary pathogen responsible for this condition, a situation aggravated by drug resistance and virulence, especially its ability to form biofilms [8].

Biofilms are inter- or intraspecific communities of bacteria maintained in a self-produced extracellular matrix by adhering to biotic or abiotic surfaces [9]. They have been associated with increased resistance to antimicrobial agents, environmental stress, and a poor host immune system, contributing to the chronicity of infections [10].

One way to minimize mastitis is through proper hygiene during milking. This is considered one of the fundamental pillars in preventing damage to the mammary gland [11]. The use of appropriate techniques, effective disinfection of the milker, and the disposal of milk from infected animals are measures that reduce the transmission of contagious pathogens, such as *Staphylococcus aureus* and *Streptococcus agalactiae* [12]. Alternative strategies, such as the use of herbal medicines and natural additives, have stood out as complementary approaches to prevention, meeting the demand for sustainable practices and reducing antibiotic use [13].

Veterinary pharmacotherapy has been highlighted in the scientific community due to the use of bioactive compounds as alternatives to synthetic drugs. Among these compounds, the pyroligneous extract derived from the condensation of gases released during the pyrolysis of Eucalyptus wood (Hybrid *Eucalyptus urophylla x Eucalyptus grandis*—clone I144) has shown significant antimicrobial potential due to its diverse chemical composition, including an abundance of bioactive compounds, such 2-methoxy-phenol, phenol, and 2,6-dimethoxy-phenol [14,15], that can protect against mastitis-causing pathogenic bacteria, potentially affecting their membrane integrity and inhibiting nucleic acid synthesis or energy metabolism [16,17]. The complexity of the chemical constituents of the pyroligneous extract suggests potential for synergistic antimicrobial effects. In this context, this study evaluates the antimicrobial and antibiofilm potential of a pyroligneous extract from *Eucalyptus urograndis* (clone I144) wood residues against mastitis-causing multidrug-resistant *Staphylococcus aureus*.

## 2. Material and Methods

### 2.1. Bacterial Isolates

The isolates used in this study were collected with the approval of the Animal Use Ethics Committee of the Federal Rural University of Pernambuco (UFRPE), under license No. 3291200922, and the Research Ethics Committee of the University of Pernambuco (UPE).

Milk samples were collected from five dairy farms located in the state of Pernambuco, Brazil. A total of 676 mammary quarter milk samples were obtained and analyzed using the California Mastitis Test (CMT). Among these, 335 samples that showed a reaction of one cross (+) or greater, or were positive in the black background cup test, were selected for microbiological analysis.

Samples were transported under refrigeration to the laboratory and processed for bacterial isolation. Samples were directly inoculated onto blood agar with 5% sheep blood and incubated at 37 °C for 24 to 48 h. After this period, isolated colonies underwent Gram staining to verify morphology, in addition to catalase, coagulase, and mannitol fermentation tests [18] (Carter, 1989). Isolates initially identified as *Staphylococcus* spp. based on colonial morphology, Gram staining, and catalase test were subsequently subjected to genomic DNA extraction, following the protocol in [19]. According to [20], DNA extracted from the isolates was quantified by spectrophotometry, using absorbance readings at 260 nm. Molecular identification of *Staphylococcus aureus* was confirmed by PCR, through the amplification of the *nuc* gene [21]. Additionally, screening for the resistance genes listed in Table 1 was also performed. The target genes included *blaZ, mecA*, and *mecC* (associated with β-lactam resistance) [22,23,24].

The PCR reaction had a final volume of 12.5 µL and contained 100 ng of genomic DNA, 0.5 µL of each primer (10 pmol), 6.25 µL of GoTaq Green Master Mix (Promega, Madison, WI, USA), and 2.5 µL of ultrapure Milli-Q water. Amplifications were done in a Peltier thermocycler (BIOER, Hangzhou, China) under the following conditions: initial denaturation at 94 °C for 5 min, followed by 37 cycles of a denaturation step at 94 °C for 1 min; primer annealing at 55 °C for 0.5 min, and extension at 72 °C for 1 min; followed by a final extension at 72 °C for 7 min [19]. Samples were then subjected to electrophoresis on a 2% agarose gel for 60 min at 100 volts. Afterwards, the gels, stained with BlueGreen, were visualized under ultraviolet light and photographed. The *S. aureus* N315 strain was used as a positive control in all reactions, and the negative control contained all the substances for the PCR reaction except the DNA. Isolates confirmed as *S. aureus* were cultured in BHI, incubated overnight, and then frozen at −80 °C with 20% glycerol, becoming part of the laboratory’s bacterial biobank.

For the current study, the isolates were reactivated by plating on mannitol salt agar (Difco Laboratories Inc., Detroit, MI, USA), followed by incubation in a bacteriological incubator at 37 °C (±1 °C) for 24 to 48 h. After this period, bacterial growth was verified, and colonies were selected based on Gram staining and the catalase test. The selected colonies were re-streaked onto fresh mannitol salt agar to obtain confluent growth. A portion of this growth was used for DNA extraction and re-confirmation using the *nuc* gene.

From the extracted samples, 10 *Staphylococcus aureus* strains were utilized for the present study.

### 2.2. Plant Material

Wood samples of the hybrid *Eucalyptus urophylla* × *Eucalyptus grandis* (clone I144) were collected from 8-year-old plantations in the experimental area of the Agricultural Sciences Unit of the Federal University of Rio Grande do Norte (05°51′30″ S and 35°21′14″ W), located in the municipality of Macaíba, state of Rio Grande do Norte, Brazil.

### 2.3. Production of Pyroligneous Extract

A pyroligneous extract was obtained from wood samples of the *Eucalyptus urograndis* (clone I144) hybrid used to produce charcoal, with a diameter ranging from 6 to 24 cm and a length of 1.0 to 3.0 m. The samples were left in the field to dry for two to six months until they reached a moisture content of 15–30%. The wood was taken to the charcoal production unit and carbonized at a heating rate of 1.25 °C/min to 2.0 °C/h, reaching a final temperature of 400–550 °C. Carbonization was conducted in masonry furnaces to produce a pyroligneous extract. The bed carbonization gases were conducted through a metallic cyclonic condenser maintained at 25 °C by cooling with running water. In this process, a portion of the smoke liquefies, producing crude carbonization liquids. These liquids are initially recovered at temperatures of 120–180 °C, and the process is completed at temperatures of 350–450 °C. The crude liquids are retrieved from ten carbonizations, each using 500 kg of wood. The crude condensed liquids were kept in a refrigerator (Consul) at 2-4 °C until further use. Subsequently, all the resulting liquids were mixed to form a composite sample. From the composite sample, ten aliquots (2–4 L) were collected and filtered using a filter set equipped with MF-Millipore membrane filters (0.22 µm porosity). Subsequently, the aliquots were tridistilled under vacuum at 1.0–25.0 mmHg at temperatures in the range of 75–100 °C, with the process interrupted at temperatures above 100–120 °C. Wood tar and oils resulting from distillation were discarded. In this step, the purified pyroligneous extract obtained was pharmaceutical grade and free of contaminants. After purification, the pyroligneous extract was kept in dark flasks under refrigeration at 2–4 °C until further use. The material obtained was free of carbamates, organochlorines, phosphorus, pyrethroids, polycyclic aromatic hydrocarbons, mycotoxins, heavy metals, dioxins, and furans.

For the GC-MS analyses, 1.5 mL of a concentrated ammonium hydroxide solution (Ammonia Solution UN 2672, Caledon, ON, Canada) was added to 5 mL aliquots of the aqueous PA samples to raise the pH from 2.5 to approximately 5. Subsequently, extractions were performed by adding 3 mL of different solvents separately: dichloromethane (Tedia, Aparecida de Goiânia, Brazil), diethyl ether, and ethyl acetate (Merck, Kenilworth, NJ, USA). All solvents were HPLC grade. Three extracts were thus produced. After liquid–liquid extraction, 1 mL of the organic fraction obtained with each solvent was transferred to a GC vial and promptly analyzed.

GC-MS analyses of the samples were performed using a Shimadzu QP 2010 system (Shimadzu, Kyoto, Japan). Separation was carried out on a CP-Wax column (Restek 52 DB, 30 m length, 0.25 mm diameter, 0.25 μm film thickness), with the injector temperature maintained at 250 °C. Chromatographic runs were performed to achieve optimal compound separation until the following analysis routine was defined. Samples (1 µL) were injected at a split ratio of 1:10, and the oven temperature program was set at 50 °C for 2 min, ramping at 2 °C min^−1^ from 50 to 240 °C, and held for 2 min. Helium was used as the carrier gas at a constant flow rate of 1 mL min^−1^. Major compounds (>20% area) and secondary compounds (~0.02%) were detected and identified by NIST library. All chemical compounds reported here showed mass spectrum similarities above 85% [14,25], as shown in Table 2. The constituents are standardized due to the same cultivation conditions, plant age, and planting area, as well as the same conditions of extract production—carbonization, condensation, and refinement [26,27]. Thus, the parameters for the chemical composition are kept consistent, as demonstrated in the composition tables obtained by GC/MS in the studies of Gama et al. (2023 and 2024) [14,15].

### 2.4. Antimicrobial Activity

#### 2.4.1. Disk Diffusion Method

The ten isolates were tested for their antimicrobial sensitivity profiles using the disk diffusion method, as described in the protocol established by the Clinical and Laboratory Standards Institute [28]. The antimicrobials tested were Penicillin G (10 µg), Ampicillin (10 µg), Cefoxitin (30 µg), Norfloxacin (10 µg), Ofloxacin (05 µg), Gentamicin (10 µg), Amikacin (30 µg), *Erythromycin* (15 µg)*,* and *Sulfazotrim* (25 µg). A standard strain of *S. aureus* ATCC 25923 was used as a positive control. The profile of resistant bacteria was determined by measuring the diameter, expressed in millimeters, on the antibiogram, which was then interpreted according to the clinical cutoff standard recommended by the Brazilian Committee for Antimicrobial Sensitivity Tests [29].

#### 2.4.2. Minimum Inhibitory Concentration (MIC)

The microdilution technique was performed to verify the minimum inhibitory concentration—MIC optical reading at 630 nm, as per the CLSI guidelines [28]. The pyroligneous extract concentrations used were 0.156%, 3.062%, 6.125%, 12.5%, 25%, and 50%. MIC was defined as the lowest concentration of pyroligneous extract that inhibits the growth of the test culture on the microplate. The content of the microplate wells was composed of 95 μL of BHI, 100 μL of each concentration of pyroligneous extract, and 5 μL of bacteria. The plates were incubated at 37 °C. All samples were observed in triplicate, and the mean ± SD values of optical density were recorded and compared between 0 and 24 h of incubation. A standard strain of *Staphylococcus aureus* (ATCC 25923) was used as a quality control, heart infusion broth (BHI) was used as a negative control, and BHI with the addition of 1% iodine (100 µg mL^−1^) was used as a positive control. The 1% iodine was selected as a positive control due to its widespread and well-established use in milking practices by rural producers. This choice reflects its relevance as a comparative standard in routine mastitis management in the field [30].

#### 2.4.3. Minimum Bactericidal Concentration (MBC)

The determination of Minimum Bactericidal Concentrations (MBCs) was performed based on the MIC of each microorganism, where the solutions from the MIC wells were streaked using a 10 μL platinum loop onto Petri dishes containing Brain Heart Infusion Agar in triplicate. The plates were incubated in an air circulation incubator at 37 ± 1 °C for 24 h. Subsequently, the presence or absence of microbial colonies was assessed.

### 2.5. Phenotypic Characterization of Biofilm Formation Using the Congo Red Agar Test

The six *S. aureus* strains used in the crystal violet assay were identified as biofilm producers through cultivation on Congo Red Agar [31] (Freeman et al., 1989). The Congo Red agar plates were inoculated and incubated at 37 °C for 24 h. The formation of rough and blackened colonies indicates the presence of bacterial biofilms, while smooth and reddish colonies indicate the absence of such biofilms. The *S. aureus* strains ATCC 25923 and ATCC 12228 were also used as positive and negative controls, respectively.

### 2.6. Biofilm Inhibition

Biofilm formation was analyzed using the crystal violet technique. Initially,100 µL of BHI, 100 µL of *Eucalyptus urograndis* (clone I144) at concentrations of 50%, 25%, 12.5%, 6.25%, and 3.13% pyroligneous extract, and 5 µL of inoculum adjusted between 0.08 and 0.1 were added to each well of a 96-well microplate, corresponding to MacFarland scale number 0.5 (density of 1.5 × 10^8^ cells per mL), and stirred for 24 h at 120 rpm in a stirring oven at 37 °C.

After 24 h, the wells were washed twice with 200 µL of 0.9% saline solution to remove non-adherent cells. They were then kept at room temperature for 1 h to allow the biofilms to set. This was followed by the addition of 200 µL of 0.2% crystal violet for 15 min. The dye was later removed by washing the wells twice with 200 µL of 0.9% saline solution and kept at room temperature for 15 min. Subsequently, 200 µL of absolute alcohol was added for 15 min to resuspend the biofilm. Afterward, the microplate was placed in a microplate reader at a wavelength of 630 nm for reading.

For biofilm inhibition determination, the mean optical density (OD) values of the resuspended biofilm were compared between the test wells and the negative control (distilled water) [32,33].

### 2.7. Statistical Analyses

Optical density (OD) data were used to determine both the minimum inhibitory concentration (MIC) and biofilm inhibition rate. Absorbance values were expressed as mean ± standard error. Data normality and homogeneity of variances were assessed using the Shapiro–Wilk and Levene’s tests, respectively, with no transformations required. Statistical analysis was performed using analysis of variance (ANOVA), followed by Tukey’s test in Sisvar software (version 5.6), adopting a significance level of 5% (*p* < 0.05). MIC was determined by comparing OD means at 0 and 24 h of incubation, selecting the lowest concentration that showed no significant difference between the time points (*p* > 0.05). For biofilm inhibition, OD means of the treatments were compared to the control after 24 h.

## 3. Results and Discussion

### 3.1. Antimicrobial Sensitivity Profile

Table 3 describes the sensitivity profile. All isolates demonstrated bacterial resistance to at least three different classes of antibiotics, including macrolides and lincosamides, fluoroquinolones, aminoglycosides, penicillins, sulfonamides, and cephalosporins (Table 2), thereby being classified as multidrug resistant (MDR) [31]. In addition to the antibiogram, all isolates were screened by PCR for the resistance genes *blaZ, mecA*, and *mecC*. No isolate carried mecC. All isolates harbored at least one genetic marker (*blaZ, mecA*, or both). Importantly, every isolate showing phenotypic resistance to cefoxitin was *mecA*-positive, which corroborates the phenotypic findings.

Related bacteria were considered multidrug resistant due to their resistance to multiple classes of antibiotics [34] (World Health Organization, 2024). All microorganisms were resistant to macrolides, penicillins, and sulfonamides, likely due to the presence of resistance genes, mutations in ribosomal proteins, the action of efflux pumps, and enzymes that inactivate antibiotics, allowing them to modify or degrade antibiotic molecules and render them ineffective. These mechanisms can act alone or in combination, conferring significant resistance to these antimicrobials [35,36].

The ability of *Staphylococcus aureus* to tolerate β-lactam antimicrobial agents can be attributed to two major biological strategies. The first is related to the synthesis of β-lactamase (penicillinase), encoded by the *blaZ* gene [37], an enzyme responsible for hydrolyzing and inactivating β-lactam molecules. The second involves the expression of a modified penicillin-binding protein (PBP2a), which is encoded by the mec genes (*mecA/mecC*) that are integrated into the chromosome within the staphylococcal cassette chromosome mec (*SCCmec*), a highly transmissible mobile genetic element (MGE) among bacterial populations. The structural conformation of this protein [38] prevents the antibiotic from accessing its catalytic site, thereby preventing target binding and consequently not interfering with bacterial cell wall biosynthesis. Thus, the microorganism maintains its viability and replication capability even in the presence of these drugs [39].

Antibiotics play a crucial role in preventing and treating bacterial infections. However, the indiscriminate and excessive use of these drugs has generated increasing concern due to the development of bacterial resistance. Still, the uncontrolled use of antibiotics puts selective pressure on *S. aureus*, resulting in the emergence of multidrug-resistant strains [40], which reduces treatment options. These antimicrobials have complex mechanisms of action and can alter binding site structures, reduce cell membrane permeability, and inactivate enzymes present in bacteria [41].

These isolated strains were resistant to β-lactams, such as penicillins and cephalosporins, likely due to the presence of the *mecA* gene, which encodes the protein PBP2a—a PBP with low affinity for β-lactam antibiotics—that allows cell wall synthesis even in the presence of these antibiotics. This gene is located in the *SCCmec* mobile genetic element, facilitating its spread between strains of *S. aureus* [42,43]. Resistance to aminoglycosides occurs by inhibiting bacterial protein synthesis as it binds with high affinity to the 16S ribosomal RNA of the 30S subunit of bacterial ribosomes, thereby interfering with the translation process. This binding causes errors in the reading of the genetic code, leading to the production of defective proteins that insert into the cytoplasmic membrane, compromising its integrity and facilitating the additional entry of aminoglycosides into the cell, amplifying the bactericidal effect [44,45]. The resistance of the strains to fluoroquinolones, such as the antibiotics norfloxacin and ofloxacin, was also observed. These antimicrobials act by inhibiting the bacterial enzyme DNA gyrase and directly acting on the enzyme topoisomerase IV in *Staphylococcus aureus*. This enzyme is essential for DNA replication, transcription, and repair. Fluoroquinolones stabilize the breaks induced by topoisomerase by binding to the enzyme-DNA complex. This prevents the rewiring of DNA strands, leading to bacterial cell death. The resistance of *Staphylococcus aureus* to fluoroquinolones primarily occurs through two mechanisms: i. mutations in the genes that encode the target enzymes of fluoroquinolones, which are encoded by the *grlA* and *grlB* genes, that are essential for bacterial DNA replication and transcription. The mutations in the quinolone resistance-determining region (QRDR) of these genes result in changes in enzymes, decreasing the affinity for fluoroquinolones [46,47,48]. ii. *S. aureus* can develop resistance to fluoroquinolones through the overexpression of efflux pumps, such as *NorA, NorB*, *NorC*, and *MepA*. These carrier proteins actively remove fluoroquinolones from the inside of the bacterial cell, reducing their intracellular concentration and, consequently, their effectiveness [49,50]. Overexpression of these pumps can occur due to mutations in promoter regions or regulatory genes, such as *mgrA* [48].

All strains were resistant to erythromycin, likely due to its long-term use as an alternative to penicillin, cephalosporins, and other beta-lactams commonly used to control mastitis caused by *Staphylococcus* [51,52]. The strains were also resistant to sulfamethoxazole-trimethoprim, possibly due to mutations in the dhps gene, resulting in an enzyme with a lower affinity for sulfonamides. In addition, the acquisition of resistance genes, such as sul1, sul2, and sul3, through mobile genetic elements also contributes to resistance to sulfonamides [53,54]. Another point to note is the continuous exposure of commensal *Staphylococcus* spp. to semisynthetic macrolide molecules, such as clarithromycin, azithromycin, and telithromycin, which are clinically used to treat infections caused by bacteria other than *S. aureus*. Thus, this contributes to the erythromycin resistance commonly found in clinical isolates of *S. aureus* [55].

*S. aureus* is recognized as a commensal, colonizing, and latent organism with a high propensity to adapt to various environments and antimicrobial agents [56]. It is also a highly contagious pathogen that induces long-lasting, chronic infections and can develop resistance to various antibiotics, representing a significant threat to public health [57,58]. Although antibiotics inhibit the growth of these bacteria, the indiscriminate use of these antimicrobials can generate resistant strains, requiring different types of control. Thus, it is necessary to develop various alternatives to combat *S. aureus*, including those relevant to animal husbandry environments, where its spread can be reduced.

Studies have demonstrated the effectiveness of pyroligneous extract as an alternative antimicrobial agent against various bacterial strains. Ref. [59] evaluated the antibacterial activity of the *Rhizophora apiculata* pyroligneous extract against *Staphylococcus aureus*, observing a significant reduction in the bacterial population after treatment due to the presence of phenolic compounds that present an antibacterial mechanism of action.

Ref. [60] conducted a systematic review on the antimicrobial potential of pyrolignous extracts, highlighting their bactericidal efficacy against pathogenic bacteria and suggesting their use as a natural antimicrobial agent in various applications. In this context, *Eucalyptus urograndis* (clone I144) pyroligneous extract was efficient as an antimicrobial compared to other pyroligneous extracts and commercial antimicrobials by acting directly on the cell wall of bacteria due to the presence of organic acids and phenolic compounds [25] as these components are essential antimicrobial agents [14].

### 3.2. In Vitro Test

In vitro test results regarding the minimum inhibitory concentration (MIC) and the minimum bactericidal concentration (MBC) demonstrated antimicrobial action at 12.5% and 25%, respectively, in all strains tested. The extract is considered bactericidal, since the MBC/MIC ratio is equal to 2 [61,62].

The bacterial strains tested exhibited a resistance profile to several conventional antimicrobials, as determined by the sensitivity test, which tested several antibiotics including beta-lactams, aminoglycosides, and sulfonamides. The strains showed sensitivity to the pyroligneous extract, indicating that the compounds present in the extract at a concentration of at least 12.5% are effective in inhibiting in vitro growth. This result corroborates the study with *Eucalyptus grandis* pyroligneous extract, which demonstrated potential as an antimicrobial agent due to the presence of phenolic compounds, organic acids, and aldehydes that confer biocidal activities [15,63,64,65]. These compounds act by destabilizing the cell membrane of microorganisms and inhibiting enzymes essential for bacterial survival. Recent studies have also demonstrated that pyroligneous extract exhibits significant inhibitory activity against *S. aureus* strains, thereby reducing bacterial biofilm formation and causing structural damage to the cell membrane [66]. The results are likely due to the antimicrobial activity of pyroligneous extract derived from eucalyptus (*Eucalyptus urograndis*) and bamboo (*Bambusa vulgaris*). Compounds such as furfural and phenols, known for their antimicrobial properties, are present in pyroligneous extracts [14], with results similar to those found in the present study.

These studies indicate that pyroligneous extract has promising antimicrobial activity against *Staphylococcus aureus*. It is essential to note that efficacy can be influenced by factors such as the pH of the extract, as reported by [67,68], who noted a decrease in antimicrobial activity with increasing pH. However, inhibitory activity remained at neutral and slightly alkaline levels.

### 3.3. Biofilm Inhibition by the Crystal Violet Technique

The *Eucalyptus urograndis* (clone I144) pyroligneous extract inhibited the formation of biofilms (*p* < 0.05) compared to the control, in the concentration range of 3.13% to 12.5%. The concentration of 12.5% inhibited 3/6 of the bacterial strains of *S. aureus* (Figure 1).

The effect of *Eucalyptus urograndis* (clone I144) pyroligneous extract on biofilms was evaluated using six biofilm-forming MDR strains in a phenotypic characterization of biofilm formation, as determined by the Congo Red Agar test. This test is widely used to detect and characterize bacterial biofilms due to Congo Red Agar’s ability to interact with extracellular matrix components, such as polysaccharides and proteins, highlighting the test’s importance as a diagnostic and research tool [69].

Strains detected as biofilm producers confer advantages, such as increased resistance to antibiotics and chemical agents, as well as evasion of the host’s immune system [70]. Thus, the verified antibiofilm capacity of pyroligneous extracts is essential to ensure a degree of efficiency of the antibacterial alternative agent. This antibiofilm activity with plant extracts occurs due to inhibition of initial bacterial adhesion to bacterial cells [71], justifying biofilm inhibition at concentrations from 3.13 to 12.5%

Pyroligneous extracts contain phenolic compounds, flavonoids, terpenoids, and alkaloids that are the main phytochemical groups associated with these actions. They act synergistically to prevent the establishment and maturation of biofilms, interfering with cell communication and the degradation of the extracellular matrix, which justifies lower biofilm production at concentrations of 12.5 to 50%. These findings are consistent with studies that used ethanolic extracts of *Ocimum sanctum* (holy basil) and *Curcuma longa*, which showed positive results in inhibiting the initial formation of biofilms produced by *Staphylococcus aureus* and *Pseudomonas aeruginosa* [72,73]. This antimicrobial effect is mainly attributed to the presence of phenolic compounds, which are responsible for inhibiting microbial growth and interfering with the formation of biofilms by these species commonly associated with persistent infections. This occurs primarily through the disruption of essential metabolic pathways, inhibiting steps in the synthesis of proteins, lipids, and nucleic acids, resulting in structural and functional alterations that lead to deficient bacterial replication [74].

The biofilm inhibition in the strains likely occurred due to the presence of organic acids in the extract, which penetrate the extracellular matrix of the biofilm and acidify the intracellular environment of the bacteria, leading to inhibition of microbial growth and destabilization of the biofilm structure [75]. This destabilizes the extracellular matrix, making them promising agents to control biofilms in various applications [69].

The lower concentrations of pyroligneous extracts (6.25% and 3.13%) of strains 02, 03, and 04 did not inhibit biofilm formation. This is likely due to metabolic dormancy in biofilms, making them less susceptible to antibiotics, which may be a response to the inactivity of low concentrations of pyroligneous extracts [76]. Since phenolic compounds, which are among the main components of the extract, typically act on active metabolic processes [16,17], it is likely that dormant cells are less sensitive to the treatment, especially when the concentrations are not sufficiently high to penetrate and affect the biofilm structure. The reduced action of pyroligneous extracts at these concentrations can be attributed to the presence of bacteria in various metabolic states, which are commonly found in areas with low oxygen and nutrient contents, making them less susceptible to the action of antibiotics, which typically target metabolically active cells. The absence of biofilm inhibition can be explained by the formation of persistent cells, such as phenotypically variant subpopulations that survive exposure to antimicrobials, such as pyroligneous extracts. These cells can reactivate the infection after treatment has ended, making biofilm eradication difficult [77].

The inefficient results can also be validated by the type of extracellular matrix that the bacteria’s biofilms produced. They likely acted as a physical barrier, restricting the penetration of antimicrobial agents and possibly decreasing the extract’s action, thereby hindering its diffusion to the bacteria present in the innermost layers of the biofilm [78]. In addition, the proximity of bacteria within biofilms facilitates the horizontal transfer of genetic material, including antimicrobial resistance genes, such as those against the antimicrobials found in the pyroligneous extract, which increases overall community resistance [79].

In this context, studies have shown that herbal extracts can interfere with the initial processes of biofilm formation, reduce the production of extracellular matrix, and increase the permeability of bacterial membranes, thereby facilitating the action of traditional antimicrobials. These properties make herbal medicines an alternative for biofilm control in clinical and industrial applications [73]. The search for therapeutic alternatives is necessary due to the resistance of bacteria in biofilms to conventional treatments. Herbal extracts are a promising strategy due to the presence of bioactive compounds, such as phenols, flavonoids, and alkaloids, which have antimicrobial and antibiofilm properties [80].

## 4. Conclusions

The results obtained in this study demonstrate that the evaluated extract exhibits significant antimicrobial activity against *Staphylococcus aureus* strains isolated from bovine mastitis cases, including multidrug-resistant variants and methicillin-resistant *S. aureus* (MRSA). Furthermore, biofilms were inhibited at different concentrations, ranging from 3.13% (1/6) and 6.25% (3/6) to 12.5% (3/6). These findings indicate that the extract represents an effective preventive strategy against mastitis-causing pathogens that are difficult to treat, making it a promising alternative for reducing dependence on synthetic antibiotics. In vivo studies are necessary to confirm these findings and provide a foundation for evidence-based clinical guidelines.

## Figures and Tables

**Figure 1 microorganisms-13-02771-f001:**
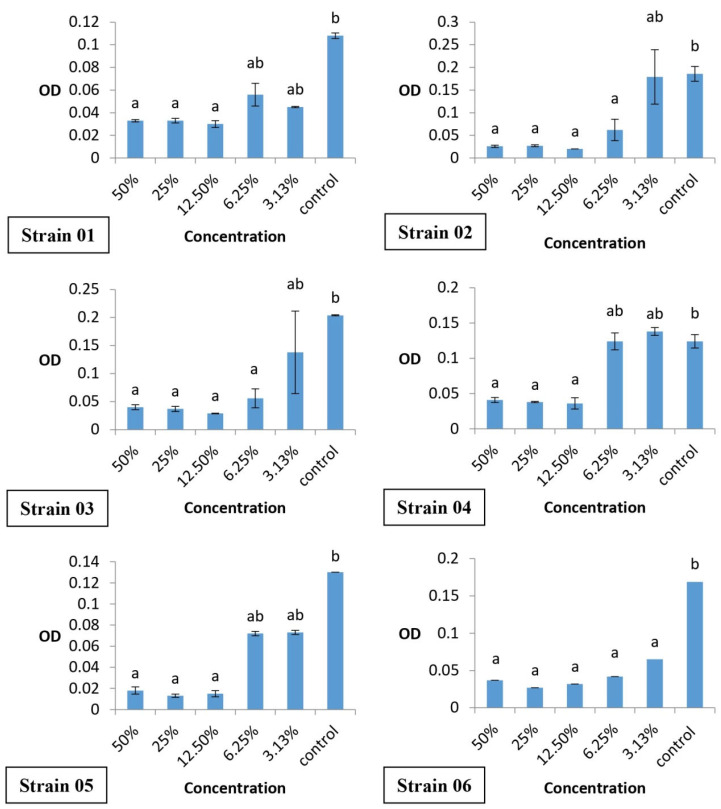
Optical density (OD) of MDR resuspended biofilms treated with different concentrations of *Eucalyptus urograndis* (clone I144) pyroligneous extract. OD values are expressed as mean ± standard error, based on six bacterial strains. Different lowercase letters indicate statistically significant differences between treatments (*p* < 0.05).

**Table 1 microorganisms-13-02771-t001:** Molecular Identification of *Staphylococcus aureus* by PCR.

Gene	Primer Sequence	Fragment Size (pb)	Reference
*nuc*	R: AGCCAAGCCTTGACGAACTAAGC	270	Kateete et al. (2010) [21]
	F: GCGATTGATGGTGATACGGTT		
*blaZ*	F: AAGAGATTTGCCTATGCTTC R: GGCAATATGATCAAGATAC	517	Sawant et al. (2009) [22]
*mecA*	F: TGGTATGTGGAAGTTAGATTGGGAT R: CTAATCTCATATGTGTTCCTGTATTGGC	155	Nakagawa et al. (2005) [23]
*mecC*	F: CATTAAAATCAGAGCGAGGC R: TGGCTGAACCCATTTTTGAT	188	Paterson et al. (2012) [24]

R: Reverse; F: Forward.

**Table 2 microorganisms-13-02771-t002:** Major GC/MS components of the pyroligneous extract of *Eucalyptus urograndis* I144.

Major Compounds	Relative Peak Areas (%)
Furfural	17.2
2-Methoxy-phenol	9.4
Phenol	8.5
5-Methyl-2-Furancarboxaldehyde	4.4
2-Methoxy-3-methyl-Phenol	4.4
1-(2-Furanyl)-ethanone	1.5
2,3-Dimethyl-2-cyclopenten-1-one	1.4
2,5-Dihydro-3,5-dimethyl-2-furanone	1.0
3-Methyl-2-Cyclopenten-1-one	2.4
1,2,3-Trimethoxy-5-methyl-benzene	1.9
3-Methyl-Phenol,	3.3
1,2,5-Trimethoxybenzene	3.5
4-Methyl-Phenol	2.3
1,2-Cyclopentanedione-3-methyl	3.3
2,6-Dimethoxy-Phenol,	10.7
3-ethyl-2-hydroxy-2-cyclopenten-1-one	1.0
4-Ethyl-2-methoxy-phenol	1.8

Adapted from Table 3 in the paper by Gama et al., 2023 [14].

**Table 3 microorganisms-13-02771-t003:** Resistance profile of the 10 *Staphylococcus aureus* isolates to different classes of antimicrobials.

Strains/Antimicrobials	PEN		CEF	AMI		FLU		MAC	SUL	Resistance Pattern	
	PEN	AMP	CFO	AMI	GEN	NOR	OFX	ERI	SUT	N	%
SA1	R	R	S	S	S	S	S	R	R	4/9	44.44%
SA2	R	R	R	R	R	S	S	R	R	7/9	77.77%
SA3	R	R	R	R	R	S	S	R	R	7/9	77.77%
SA4	R	R	R	R	R	S	S	R	R	7/9	77.77%
SA5	R	R	R	R	R	S	S	R	R	7/9	77.77%
SA6	R	R	R	R	R	S	S	R	R	7/9	77.77%
SA7	R	R	R	R	R	S	S	R	R	7/9	77.77%
SA8	R	R	R	R	R	R	R	R	R	9/9	100%
SA9	R	R	R	R	R	S	S	R	R	7/9	77.77%
SA10	R	R	R	R	R	S	S	R	R	7/9	77.77%

PEN—Penicillins (Penicillin 10 µg and Ampicillin 10 µg); CEF—Cephalosporins (Cefaxotin 30 µg); AMI—Aminoglycospides (Amikacin 30 µg and Gentamicin 10 µg); FLU—Fluoroquinolones (Norfloxacin 10 µg and Ofloxacin 05 µg); MAC—Macrolides and Lincosamides (Erythromycin 15 µg); SUL—Sulfonamide (Sulfazotrim 25 µg); N—number; %—percentage.

## Data Availability

The original contributions presented in this study are included in the article. Further inquiries can be directed to the corresponding author.

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
