# Peer review of "Evaluation of Antimicrobial and Antibiofilm Activity of Eucalyptus urograndis (Clone I144) Pyroligneous Extract on Bovine Mastitis Isolate of Multiple-Drug-Resistant Staphylococcus aureus Strains"

_microorganisms, 2025, doi:10.3390/microorganisms13122771_

Round 1
Reviewer 1 Report
Comments and Suggestions for Authors
Previously published articles (Gama et al., 2023) contained the results of a study on the antimicrobial activity and chemical profile of wood vinegar from eucalyptus (E. urophylla × E. grandis). Like the authors of the peer-reviewed article, Gama et al. used Staphylococcus aureus bacteria as the object of study, as well as a number of the same methods. It would be desirable for the authors to examine in more detail the similarities and differences between their data and those reported by Gama et al., and also to justify the novelty of their work.
The main shortcoming of the manuscript is that the obtained data are described in short sentences or references to the work of other authors, rather than in figures or tables. For example, the authors write that all chemical compounds reported here showed mass spectrum similarity above 85% [20, 21]. The obtained data should be presented in the form of tables or figures.
- There is a discrepancy between the volume of data obtained by the authors and the length of the manuscript The manuscript can be significantly shortened and presented as a short report.
[14] Gama, GSP; Pimenta, AS; Feijó, F; Santos, C; Fernandes, B; Oliveira, M; Souza, E; Monteiro, T; Fasciotti, M; Azevedo, T. Antimicrobial activity and chemical profile of wood vinegar from eucalyptus (E. urophylla × E. grandis – clone I144) and bamboo (Bambusa vulgaris). World J Microbiol Biotechnol 2023, 39, 186. https://doi.org/10.1007/s11274-023-03628-x
[20] Gama, GSP; Pimenta, AS; Feijó, FMC; Santos, CS; Castro, RVO; Azevedo, TKB; Medeiros, LCD. Effect of pH on the antibacterial and antifungal activity of wood vinegar (pyroligneous extract) from eucalyptus. Rev Árvore 2023, 47. https://doi.org/10.1590/1806-468 908820230000011 469
[21] Pimenta, AS; Fasciotti, M; Monteiro, TVC; Lima, KMG. Chemical composition of pyroligneous acid obtained from Eucalyptus GG100 clone. 470 Molecules 2018, 23, 426. https://doi.org/10.3390/molecules23020426
Author Response
Dear Reviewer
We thank the reviewer for the comments. We agree with this comment and the authors explain the changes in the text. And a table was added for a better understanding of the data. Major GC/MS components of the pyroligneous extract of Eucalyptus urograndis I144. We add major GC/MS components of the pyroligneous extract of Eucalyptus urograndis I144.
Comments 1: Previously published articles (Gama et al., 2023) contained the results of a study on the antimicrobial activity and chemical profile of wood vinegar from eucalyptus (E. urophylla × E. grandis). Like the authors of the peer-reviewed article, Gama et al. used Staphylococcus aureus bacteria as the object of study, as well as a number of the same methods. It would be desirable for the authors to examine in more detail the similarities and differences between their data and those reported by Gama et al., and also to justify the novelty of their work.
[14] Gama, GSP; Pimenta, AS; Feijó, F; Santos, C; Fernandes, B; Oliveira, M; Souza, E; Monteiro, T; Fasciotti, M; Azevedo, T. Antimicrobial activity and chemical profile of wood vinegar from eucalyptus (E. urophylla × E. grandis – clone I144) and bamboo (Bambusa vulgaris). World J Microbiol Biotechnol 2023, 39, 186.
Response 1: We thank the reviewer for the comments. We agree with this comment and the authors explain the similarities and the novelty in the following text. The article published by Gama et al. (2023) evaluates the antimicrobial activity against bacterial strains and a fungal strain, most of which are ATCC standard, except for one clinical strain of Streptococcus agalactiae. The present study differs by testing the antimicrobial activity of pyroligneous extract against clinical strains of multidrug-resistant (MDR) Staphylococcus aureus isolated from bovine mastitis. The importance of this in vitro study with this focus is to analyze possibilities for the future use of the extract as an alternative in the treatment and/or prevention of the disease. Both the study by Gama et al. and ours used the microdilution technique to determine the Minimum Inhibitory Concentration (MIC) and Minimum Bactericidal Concentration (MBC). However, the present study includes the evaluation of the antibiofilm activity of the extract. For this, the MDR S. aureus strains were assessed for biofilm production on Congo Red agar and subsequently subjected to the biofilm inhibition assay using crystal violet (item 2.5 e 2.6, line 218-241). This step allows for understanding another relevant aspect of the extract, which may contribute to its clinical use in mastitis, since interference with bacterial adhesion and biofilm formation can reduce the protection this structure provides to microorganisms against antibiotics and consequently contribute to therapeutic efficacy. And it was discussed in item 3.4. Biofilm inhibition by the crystal violet technique – line 377.
Comments 2: The main shortcoming of the manuscript is that the obtained data are described in short sentences or references to the work of other authors, rather than in figures or tables. For example, the authors write that all chemical compounds reported here showed mass spectrum similarity above 85% [20, 21]. The obtained data should be presented in the form of tables or figures.
[20] Gama, GSP; Pimenta, AS; Feijó, FMC; Santos, CS; Castro, RVO; Azevedo, TKB; Medeiros, LCD. Effect of pH on the antibacterial and antifungal activity of wood vinegar (pyroligneous extract) from eucalyptus. Rev Árvore 2023, 47. https://doi.org/10.1590/1806-468 908820230000011 469
[21] Pimenta, AS; Fasciotti, M; Monteiro, TVC; Lima, KMG. Chemical composition of pyroligneous acid obtained from Eucalyptus GG100 clone. 470 Molecules 2018, 23, 426. https://doi.org/10.3390/molecules23020426
Response 2: We thank the reviewer for the comments. We agree with this comment and the authors explain the paper did not determine the components of the pyroligneous extract. These components were determined by Gama et al., 2023. The table 2 was inserted in section 2.3. page 5, lines 182-184.
Table 2. Major GC/MS components of the pyroligneous extract of Eucalyptus urograndis I144
|
Major components |
Relative Peak Areas (%) |
|
Furfural |
17.2 |
|
2-Methoxy-phenol |
9.4 |
|
Phenol |
8.5 |
|
5-Methyl-2-Furancarboxaldehyde |
4.4 |
|
2-Methoxy-3-methyl- Phenol |
4.4 |
|
1-(2-Furanyl)-ethanone |
1.5 |
|
2,3-Dimethyl-2-cyclopenten-1-one |
1.4 |
|
2,5-Dihydro-3,5-dimethyl-2-furanone |
1.0 |
|
3-Methyl-2-Cyclopenten-1-one |
2.4 |
|
1,2,3-Trimethoxy-5-methyl-benzene |
1.9 |
|
3-Methyl- Phenol, |
3.3 |
|
1,2,5-Trimethoxybenzene |
3.5 |
|
4-Methyl- Phenol |
2.3 |
|
1,2-Cyclopentanedione-3-methyl |
3.3 |
|
2,6-Dimethoxy- Phenol, |
10.7 |
|
3-ethyl-2-hydroxy-2-cyclopenten-1-one |
1.0 |
|
4-Ethyl-2-methoxy-phenol |
1.8 |
* adapated from table 3 of the paper by Gama et al., 2023
Gama, GSP., Pimenta, AS., Feijó, FMC. et al. Antimicrobial activity and chemical profile of wood vinegar from eucalyptus (Eucalyptus urophylla x Eucalyptus grandis - clone I144) and bamboo (Bambusa vulgaris). World J Microbiol Biotechnol 39, 186 (2023). https://doi.org/10.1007/s11274-023-03628-x
Comments 3: There is a discrepancy between the volume of data obtained by the authors and the length of the manuscript The manuscript can be significantly shortened and presented as a short report.
Response 3: We agree with this comment, but the arguments presented are necessary for the readers' understanding of the text for the steps described in the methodology, section 2, line 83 – 242.
1.Bacterial isolates
Milk samples were collected from e dairy farms located in the state of Pernambuco, Brazil. From the extracted samples, 10 Staphylococcus aureus strains were utilized for the present study.
2.Plant material
Wood samples of the hybrid Eucalyptus urophylla x Eucalyptus grandis (clone I144) were collected from 8-year-old plantations in the experimental area of the Agricultural Sciences Unit of the Federal University of Rio Grande do Norte (05° 51′ 30″ S and 35° 21′ 14″ W), located in the municipality of Macaíba, state of Rio Grande do Norte, Brazil.
3.Production of pyroligneous extract
The pyroligneous extract was obtained from wood samples of the Eucalyptus urograndis (clone I144) hybrid used to produce charcoal. For the GC-MS analyses, The compounds related in the pyroligneous extract are described.
- Antimicrobial activity
4.1 Disk diffusion method
The ten isolates were tested for their antimicrobial sensitivity profiles using the disk diffusion method, as described in the protocol established by the Clinical and Laboratory Standards Institute (CLSI)
4.2. Minimum Inhibitory Concentration (MIC) The microdilution technique was performed to verify the minimum inhibitory concentration - MIC´s optical reading at 630 nm, as per the CLSI guidelines
4.3. Minimum Bactericidal Concentration (MBC)
The determination of Minimum Bactericidal Concentrations (MBC) was performed based on the MIC of each microorganism.
4.4 Phenotypic characterization of biofilm formation using the Congo Red Agar test
The six S. aureus strains used in the crystal violet assay were identified as biofilm producers through cultivation on Congo Red Agar.
4.5 Biofilm formation
Biofilm formation was analyzed using the crystal violet technique. Initially,100 µl of BHI, 100 µl of Eucalyptus urograndis (clone I144) at concentrations of 50%, 25%, 12.5%, 6.25%, and 3.13% pyroligneous extract.
And the arguments described in the results and discussion in section 3.1. Antimicrobial sensitivity profile, line 252, 3.2. In vitro test, line 335, and 3.3 Biofilm inhibition by the crystal violet technique, line 360, are important, as these points form a discussion on sensitivity, in vitro testing, and biofilm inhibition due to pyroligneous extract. Therefore, we ask the reviewer to reconsider this contribution to the scientific community in the field of bacteriology.
Thanks
Reviewer 2 Report
Comments and Suggestions for Authors
Minor comments:
The manuscript investigates the antimicrobial and antibiofilm efficacy of Eucalyptus urograndis (Clone I144) pyroligneous extract against multidrug-resistant Staphylococcus aureus isolates from bovine mastitis. The study is scientifically relevant and addresses an important issue in sustainable veterinary medicine. However, it lacks detailed chemical characterization, molecular confirmation of resistance, and cytotoxicity evaluation. The methodology for extract standardization and statistical validation also requires clarification. Overall, the work is promising but needs substantial experimental and analytical reinforcement before publication.
[Lines 20–33]
The abstract reports antibiofilm inhibition at 3.13% concentration. However, it is not indicated whether these concentrations were selected based on prior optimization studies or cytotoxicity assessments. Could the authors clarify the rationale behind the concentration range tested?
[Lines 73–80]
The introduction attributes the extract’s antimicrobial potential to the presence of phenolic derivatives. Can the authors specify which major phenolic constituents were identified and discuss their possible modes of antibacterial or antibiofilm action?
[Lines 87–127]
The identification of Staphylococcus aureus isolates was confirmed by nuc-gene PCR; however, multidrug resistance classification appears to rely only on antibiogram data. Were any molecular markers of resistance (e.g., mecA, blaZ, erm) evaluated to strengthen the MDR characterization?
[Lines 133–156]
The preparation of pyroligneous extract involved high-temperature carbonization. Given the variability inherent in pyrolysis, how did the authors ensure reproducibility of chemical composition across batches, and were yield or compositional consistency parameters reported?
[Lines 165–175]
The GC–MS analysis section indicates detection of several compounds. Please provide quantitative or semi-quantitative data (e.g., relative peak area %) for key antimicrobial constituents such as acetic acid, phenol, cresols, and guaiacol to correlate composition with bioactivity.
[Lines 187–201]
The MIC assay employed iodine as a positive control. Considering that iodine acts through oxidative stress mechanisms distinct from phenolic compounds, how do the authors justify this comparator for interpreting the antimicrobial performance of the extract?
[Lines 233–240]
The statistical analysis utilized ANOVA followed by Tukey’s test. Were assumptions of normality and homogeneity of variances verified prior to applying these parametric tests, and were any data transformations performed if deviations were observed?
[Lines 243–254]
The authors classify all isolates as multidrug-resistant. Was methicillin resistance specifically verified using cefoxitin disk diffusion or mecA PCR assays to confirm MRSA status among the isolates?
[Lines 325–332]
The MIC and MBC values were reported as 12.5% and 25%, respectively. Did the authors determine the MBC/MIC ratio to classify the extract as bactericidal (≤4) or bacteriostatic (>4) according to standard criteria?
[Lines 350–353]
Biofilm inhibition was assessed using crystal violet staining. To corroborate these findings, were any complementary analyses such as confocal laser scanning microscopy (CLSM), scanning electron microscopy, or viable cell enumeration conducted?
[Lines 385–395]
The manuscript attributes reduced antibiofilm activity at lower concentrations to bacterial metabolic dormancy. Was this hypothesis experimentally validated, for instance, through viability assays or metabolic activity staining (e.g., resazurin or live/dead assays)?
[Lines 403–410]
The discussion compares the efficacy of Eucalyptus urograndis extract with other herbal preparations (e.g., Ocimum sanctum, Curcuma longa). Could the authors provide comparative data or literature-based discussion on the relative phenolic content and antibiofilm potency among these extracts?
[Lines 412–417]
The conclusion acknowledges the need for in vivo validation. Do the authors plan to perform cytotoxicity testing on mammalian cell lines or animal model evaluations to confirm the biocompatibility and therapeutic applicability of the pyroligneous extract?
Author Response
Dear Reviewer
We appreciate the review of the article.
Comments 1: The manuscript investigates the antimicrobial and antibiofilm efficacy of Eucalyptus urograndis (Clone I144) pyroligneous extract against multidrug-resistant Staphylococcus aureus isolates from bovine mastitis. The study is scientifically relevant and addresses an important issue in sustainable veterinary medicine. However, it lacks detailed chemical characterization, molecular confirmation of resistance, and cytotoxicity evaluation. The methodology for extract standardization and statistical validation also requires clarification. Overall, the work is promising but needs substantial experimental and analytical reinforcement before publication.
[Lines 20–33]
The abstract reports antibiofilm inhibition at 3.13% concentration. However, it is not indicated whether these concentrations were selected based on prior optimization studies or cytotoxicity assessments. Could the authors clarify the rationale behind the concentration range tested
Response 1: The concentrations used in the biofilm inhibition assay were defined based on the Minimum Inhibitory Concentration (MIC) determined in previous studies conducted with both clinical and standard strains, which showed a value of 12.5%. Based on this result, concentrations corresponding to two and four times above and below the MIC were tested, allowing the evaluation of the bacterial response at different levels of exposure to the extract.
In this way, the text in the summary was modified, page 1, line 27 - antibiofilm activity starting
Comments 2: [Lines 73–80]
The introduction attributes the extract’s antimicrobial potential to the presence of phenolic derivatives. Can the authors specify which major phenolic constituents were identified and discuss their possible modes of antibacterial or antibiofilm action?
Response 2: Changes were included. In page 2, lines 75-78, it was added compounds
2-methoxy-phenol, phenol, and 2,6-dimethoxy-phenol [14,15], and the action mechanism - … against mastitis-causing pathogenic bacteria, potentially affecting membrane integrity and inhibiting nucleic acid synthesis or energy metabolism [16,17].
Comments 3: [Lines 87–127]
The identification of Staphylococcus aureus isolates was confirmed by nuc-gene PCR; however, multidrug resistance classification appears to rely only on antibiogram data. Were any molecular markers of resistance (e.g., mecA, blaZ, erm) evaluated to strengthen the MDR characterization?
Response 3: Thank you for the comment. In our study, multidrug resistance (MDR) was defined according to Magiorakos et al. (2012) and Sweeney et al. (2018), non-susceptibility to at least one agent in three or more antimicrobial classes, based on phenotypic susceptibility testing. In addition to the antibiogram, all isolates were screened by PCR for the resistance genes blaZ, mecA, and mecC. No isolate carried mecC. All isolates harbored at least one genetic marker (blaZ, mecA, or both). Importantly, every isolate showing phenotypic resistance to cefoxitin was mecA-positive, which corroborates the phenotypic findings.
Genomic DNA was extracted from bacterial colonies using the thermal lysis method described by Fan et al. (1995). DNA concentration and purity were assessed spectrophotometrically at 260 nm (Thermo Fisher Scientific, Massachusetts, USA), following Brakstad, Aasbakk & Maeland (1992). Polymerase chain reaction (PCR) was used to amplify the resistance genes listed in Table 1. The target genes included blaZ, mecA, and mecC (associated with β-lactam resistance).
|
Gene |
Primer sequences (5’→3’) |
Amplicon (bp) |
Reference |
|
blaZ |
F: AAGAGATTTGCCTATGCTTC R: GGCAATATGATCAAGATAC |
517 |
Sawant et al. (2009) |
|
mecA |
F: TGGTATGTGGAAGTTAGATTGGGAT R: CTAATCTCATATGTGTTCCTGTATTGGC |
155 |
Nakagawa et al. (2005) |
|
mecC |
F: CATTAAAATCAGAGCGAGGC R: TGGCTGAACCCATTTTTGAT |
188 |
Paterson et al. (2012) |
The data were entered in Table 01 on the page, lines 107–108, and described in the text on lines 103–105.
The results for the blaZ, mecA, and mecC genes were reported in the text – page 7, lines 259-263.
A text was added to the discussion about these genes in page 8, lines 277-287.
Brakstad OG, Aasbakk K, Maeland JA (1992) Detection of Staphylococcus aureus by polymerase chain reaction amplification of the nuc gene. Journal of Clinical Microbiology 30:1654–1660. https://doi.org/10.1128/jcm.30.7.1654-1660.1992
Fan HH, Kleven SH, Jackwood MW (1995) Application of polymerase chain reaction with arbitrary primers to strain identification of Mycoplasma gallisepticum. Avian Diseases 39:729–735
Magiorakos AP, Srinivasan A, Carey RB, Carmeli Y, Falagas ME, Giske CG, Harbarth S, Hindler JF, Kahlmeter G, Olsson-Liljequist B, Paterson DL, Rice LB, Stelling J, Struelens MJ, Vatopoulos A, Weber JT, Monnet DL (2012) Multidrug-resistant, extensively drug-resistant and pandrug-resistant bacteria: An international expert proposal for interim standard definitions for acquired resistance. Clin Microbiol Infect 18(3):268–281. https://doi.org/10.1111/j.1469-0691.2011.03570.x
Nakagawa S, Taneike I, Mimura D, et al (2005) Gene sequences and specific detection for Panton-Valentine leukocidin. Biochemical and Biophysical Research Communications 328:995–1002. https://doi.org/10.1016/j.bbrc.2005.01.054
Paterson GK, Larsen AR, Robb A, et al (2012) The newly described mecA homologue, mecALGA251, is present in methicillin-resistant Staphylococcus aureus isolates from a diverse range of host species. Journal of Antimicrobial Chemotherapy 67:2809–2813. https://doi.org/10.1093/jac/dks329
Sawant AA, Gillespie BE, Oliver SP (2009) Antimicrobial susceptibility of coagulase-negative Staphylococcus species isolated from bovine milk. Veterinary Microbiology 134:73–81. https://doi.org/10.1016/j.vetmic.2008.09.006
Sweeney MT, Lubbers BV, Schwarz S, Watts JL (2019) Applying definitions for multidrug resistance, extensive drug resistance and pandrug resistance to clinically significant livestock and companion animal bacterial pathogens—authors’ response. Journal of Antimicrobial Chemotherapy 74:536–537. https://doi.org/10.1093/jac/dky470
Comments 4: [Lines 133–156]
The preparation of pyroligneous extract involved high-temperature carbonization. Given the variability inherent in pyrolysis, how did the authors ensure reproducibility of chemical composition across batches, and were yield or compositional consistency parameters reported?
Response 4: During the production of the pyroligneous extract, physical and chemical parameters are maintained to ensure the standardization and quality of the final product. The plant specimens used are clones of Eucalyptus urograndis (clone I144), of similar age and cultivated in plantations of the Agricultural Sciences Unit, Federal University of Rio Grande do Norte (05° 51′ 30″ S and 35° 21′ 14″ W), municipality of Macaíba, Rio Grande do Norte State, Brazil. The amount and cutting of wood portions are standardized according to Carneiro et al. (2013), as well as the processes of carbonization; condensation and decantation of the crude extract; and sequential vacuum distillation for refinement and production of the final standardized extract, which are maintained as described by Gama et al. (2023) and Pimenta et al. (2023). Thus, the parameters of chemical composition are kept consistent, as demonstrated in the composition tables obtained by GC/MS in the studies of Gama et al. (2023 and 2024).
It was added in paper, page 5, line 176 – 181 – The constituents are standardized due to the same cultivation conditions, plant age, and planting area, as well as the conditions of extract production - carbonization, condensation, and refinement [ Carneiro et al., 2013, Pimenta et al., 2023]. Thus, the parameters of chemical composition are kept consistent, as demonstrated in the composition tables obtained by GC/MS in the studies of Gama et al. (2023 and 2024).
Carneiro ACO, Santos RC, Castro RVO, Castro AFNM, Pimenta AS, Pinto EM, Alves ICN (2013) Estudo da decomposição térmica da madeira de oito espécies da região do Seridó. Rio Grande Do Norte. Revista Árvore, 37(6):20–33. https://doi.org/10.1590/S0100-67622013000600017
Gama, GSP., Pimenta, AS., Feijó, FMC. et al. Antimicrobial activity and chemical profile of wood vinegar from eucalyptus (Eucalyptus urophylla x Eucalyptus grandis - clone I144) and bamboo (Bambusa vulgaris). World J Microbiol Biotechnol 39, 186 (2023). https://doi.org/10.1007/s11274-023-03628-x
Gama GSP, Pimenta AS, Feijó FMC, Aires CAM, de Melo RR, Dos Santos CS, de Medeiros LCD, da Costa Monteiro TV, Fasciotti M, de Medeiros PL, de Morais MRM, de Azevedo TKB. Antimicrobial Impact of Wood Vinegar Produced Through Co-Pyrolysis of Eucalyptus Wood and Aromatic Herbs. Antibiotics (Basel). 2024 Nov 6;13(11):1056. doi: 10.3390/antibiotics13111056. PMID: 39596750; PMCID: PMC11590886.
Pimenta, A.S.; Gama, G.S.P.; Feijó, F.M.C.; Braga, R.M.; de Azevedo, T.K.B.; de Melo, R.R.; de Oliveira Miranda, N.; de Andrade, G.S. Wood Vinegar from Slow Pyrolysis of Eucalyptus Wood: Assessment of Removing Contaminants by Sequential Vacuum Distillation. Forests 2023, 14, 2414. https://doi.org/10.3390/f14122414
Comments 5: [Lines 165–175]
The GC–MS analysis section indicates detection of several compounds. Please provide quantitative or semi-quantitative data (e.g., relative peak area %) for key antimicrobial constituents such as acetic acid, phenol, cresols, and guaiacol to correlate composition with bioactivity.
Response 5: We thank the reviewer for the comments. We agree with this comment and the authors explain the paper did not determine the components of the pyroligneous extract. These components were determined by Gama et al., 2023. The table 2 was inserted in section 2.3. page 5, lines 182-184.
Table 2. Major GC/MS components of the pyroligneous extract of Eucalyptus urograndis I144
|
Compostos majoritários |
Relative Peak Areas (%) |
|
Furfural |
17.2 |
|
2-Methoxy-phenol |
9.4 |
|
Phenol |
8.5 |
|
5-Methyl-2-Furancarboxaldehyde |
4.4 |
|
2-Methoxy-3-methyl- Phenol |
4.4 |
|
1-(2-Furanyl)-ethanone |
1.5 |
|
2,3-Dimethyl-2-cyclopenten-1-one |
1.4 |
|
2,5-Dihydro-3,5-dimethyl-2-furanone |
1.0 |
|
3-Methyl-2-Cyclopenten-1-one |
2.4 |
|
1,2,3-Trimethoxy-5-methyl-benzene |
1.9 |
|
3-Methyl- Phenol, |
3.3 |
|
1,2,5-Trimethoxybenzene |
3.5 |
|
4-Methyl- Phenol |
2.3 |
|
1,2-Cyclopentanedione-3-methyl |
3.3 |
|
2,6-Dimethoxy- Phenol, |
10.7 |
|
3-ethyl-2-hydroxy-2-cyclopenten-1-one |
1.0 |
|
4-Ethyl-2-methoxy-phenol |
1.8 |
* adapated from table 3 of the paper by Gama et al., 2023
Gama, GSP., Pimenta, AS., Feijó, FMC. et al. Antimicrobial activity and chemical profile of wood vinegar from eucalyptus (Eucalyptus urophylla x Eucalyptus grandis - clone I144) and bamboo (Bambusa vulgaris). World J Microbiol Biotechnol 39, 186 (2023). https://doi.org/10.1007/s11274-023-03628-x
Comments 6: [Lines 187–201]
The MIC assay employed iodine as a positive control. Considering that iodine acts through oxidative stress mechanisms distinct from phenolic compounds, how do the authors justify this comparator for interpreting the antimicrobial performance of the extract?
Response 6: The study aims to evaluate the antimicrobial and antibiofilm activity of the pyroligneous extract against bacterial strains of clinical relevance to bovine mastitis. This infection can cause several sanitary and economic losses in milk production. Therefore, various efforts are made to prevent this disease, ranging from hygiene measures to the use of antiseptic substances. In this context, iodine is commonly used as an antiseptic after milking, in a step called post-dipping. For this procedure, the teat is immersed in an iodine- and glycerin-based solution to eliminate or inhibit bacterial growth on the skin surface and the teat sphincter, which may remain open for 20 to 30 minutes after milking, facilitating the entry of bacteria into the mammary gland. Thus, the choice of iodine as a positive control stems from the comparison of the extract with a substance already used in mastitis prevention, which may encourage future studies on its potential use as an alternative.
Comments 7: [Lines 233–240]
The statistical analysis utilized ANOVA followed by Tukey’s test. Were assumptions of normality and homogeneity of variances verified prior to applying these parametric tests, and were any data transformations performed if deviations were observed?
Response 7: We agreed with the suggestion and corrected the text of the statistical analysis, which was added to the page 7, lines 246-249 - Data normality and homogeneity of variances were assessed using the Shapiro–Wilk and Levene’s tests, respectively, with no transformations required. Statistical analysis was performed using analysis of variance (ANOVA), followed by Tukey’s test in Sisvar software (version 5.6), adopting a significance level of 5% (p < 0.05).
Comments 8: [Lines 243–254]
The authors classify all isolates as multidrug-resistant. Was methicillin resistance specifically verified using cefoxitin disk diffusion or mecA PCR assays to confirm MRSA status among the isolates?
Response 8: Yes. Methicillin resistance was first screened phenotypically using cefoxitin disk diffusion. Every isolate categorized as MRSA by the antibiogram was subsequently confirmed by PCR detection of mecA. The mecC gene was not detected in any isolate.
The results for the blaZ, mecA, and mecC genes were reported in the text – page 7, lines 259-263.
Comments 9: [Lines 325–332]
The MIC and MBC values were reported as 12.5% and 25%, respectively. Did the authors determine the MBC/MIC ratio to classify the extract as bactericidal (≤4) or bacteriostatic (>4) according to standard criteria?
Response 9: Thank you for pointing that out. The extract is considered bactericidal, since the MBC/MIC ratio is equal to 2 (Wald-Dickler, Holtom, Spellberg, 2018, Batista et al., 2025).
This information was included in the manuscript - page 9, lines 356-357.
Wald-Dickler N, Holtom P, Spellberg B. Busting the Myth of "Static vs Cidal": A Systemic Literature Review. Clin Infect Dis. 2018 Apr 17;66(9):1470-1474. doi: 10.1093/cid/cix1127. PMID: 29293890; PMCID: PMC5905615.
Batista, S.; Fernández-Pittol, M.; San Nicolás, L.; Martínez, D.; Narváez, S.; Espasa, M.; Garcia Losilla, E.; Rubio, M.; Garrigo, M.; Tudó, G.; et al. Design and Validation of a Simplified Method to Determine Minimum Bactericidal Concentration in Nontuberculous Mycobacteria. Antibiotics 2025, 14, 381. https://doi.org/10.3390/antibiotics14040381.
Comments 10: [Lines 350–353]
Biofilm inhibition was assessed using crystal violet staining. To corroborate these findings, were any complementary analyses such as confocal laser scanning microscopy (CLSM), scanning electron microscopy, or viable cell enumeration conducted?
Response 10: Thank you for pointing this out. We would like to justify the choice of the colorimetric assay using crystal violet as the method for evaluating biofilm inhibition. This assay enables the quantification of the total biofilm biomass (cells and matrix) and is commonly used in studies assessing plant-derived compounds with this potential (Mirzaei et al., 2022; Altuwaijri et al., 2025). The quantification of crystal violet has proven to be extremely useful as a cellular estimate of biofilm growth, being a relatively easy, reproducible assay that allows the analysis of multiple samples simultaneously and rapidly (Wilson et al., 2017). These authors also emphasize that the use of a standardized protocol, as well as a control for normalization—both applied in the present study—can eliminate methodological variability. Furthermore, a recent study published in Microorganisms (Kim et al., 2023) also employed the CV assay as the sole technique for biofilm evaluation.
Mirzaei A, Nasr Esfahani B, Ghanadian M, Moghim S. Alhagi maurorum extract modulates quorum sensing genes and biofilm formation in Proteus mirabilis. Sci Rep. 2022 Aug 17;12(1):13992. doi: 10.1038/s41598-022-18362-x. PMID: 35978046; PMCID: PMC9385855.
Altuwaijri, N.; Fitaihi, R.; Alkathiri, F.A.; Bukhari, S.I.; Altalal, A.M.; Alsalhi, A.; Alsulaiman, L.; Alomran, A.O.; Aldosari, N.S.; Alqhafi, S.A.; et al. Assessing the Antibacterial Potential and Biofilm Inhibition Capability of Atorvastatin-Loaded Nanostructured Lipid Carriers via Crystal Violet Assay. Pharmaceuticals 2025, 18, 417. https://doi.org/10.3390/ph18030417
Wilson C, Lukowicz R, Merchant S, Valquier-Flynn H, Caballero J, Sandoval J, Okuom M, Huber C, Brooks TD, Wilson E, Clement B, Wentworth CD, Holmes AE. Quantitative and Qualitative Assessment Methods for Biofilm Growth: A Mini-review. Res Rev J Eng Technol. 2017 Dec;6(4):http://www.rroij.com/open-access/quantitative-and-qualitative-assessment-methods-for-biofilm-growth-a-minireview-.pdf. Epub 2017 Oct 24. PMID: 30214915; PMCID: PMC6133255.
Kim B, Gurung S, Han SR, Lee JH, Oh TJ. Comparative Genomic Analysis of Biofilm-Forming Polar Microbacterium sp. Strains PAMC22086 and PAMC21962 Isolated from Extreme Habitats. Microorganisms. 2023 Jul 5;11(7):1757. doi: 10.3390/microorganisms11071757. PMID: 37512929; PMCID: PMC10384088.
Comments 11: [Lines 385–395]
The manuscript attributes reduced antibiofilm activity at lower concentrations to bacterial metabolic dormancy. Was this hypothesis experimentally validated, for instance, through viability assays or metabolic activity staining (e.g., resazurin or live/dead assays)?
Response 11: We sincerely appreciate the comment and understand the concern regarding the hypothesis of metabolic dormancy. In the manuscript, we mentioned this possibility because we observed that the lower concentrations of the pyroligneous extract showed reduced antibiofilm activity, which may be related to a state of bacterial dormancy—a condition characterized by a natural decrease in metabolic activity. Since phenolic compounds, which are among the main components of the extract, typically act on active metabolic processes (Alves et al., 2023; Salim et al., 2023), it is likely that dormant cells are less sensitive to the treatment, especially when the concentrations are not sufficiently high to penetrate and affect the biofilm structure. Thus, we presented this explanation only as a plausible hypothesis based on what is described in the literature.
The text about this point was added to the discussion on the page 11, lines 422-426.
Alves, BAS; Pimenta, AS; Feijó, F; Santos, C; Fernandes, B; Oliveira, M; Souza, E; Monteiro, T; Fasciotti, M; Azevedo, T. Use of product based on wood vinegar of Eucalyptus clone I144 used in the control of bovine mastitis. Vet Microbiol 2023, 279, 109602. https://doi.org/10.1016/j.vetmic.2022.109602
Salim A, Deiana P, Fancello F, Molinu MG, Santona M, Zara S. Antimicrobial and antibiofilm activities of pomegranate peel phenolic compounds: varietal screening through a multivariate approach. J Bioresour Bioprod. 2023;8(2):146-61. doi:10.1016/j.jobab.2023.01.006
Comments 12: [Lines 403–410]
The discussion compares the efficacy of Eucalyptus urograndis extract with other herbal preparations (e.g., Ocimum sanctum, Curcuma longa). Could the authors provide comparative data or literature-based discussion on the relative phenolic content and antibiofilm potency among these extracts?
Response 12: We appreciate the suggestion. We justify that phenolic compounds, such as flavonoids, present in the extracts of Ocimum sanctum, Curcuma longa, and the pyroligneous extract of Eucalyptus urograndis (clone I144), have the ability to deplete the available oxygen necessary for respiratory or oxidation pathways, inhibit protein synthesis required for bacterial cell replication, and exert antioxidant effects on DNA, RNA, protein, and lipid metabolic pathways, thereby directly interfering with biofilm production (Akinduti et al., 2024). Phenol, in particular, has the ability to inhibit dihydrofolate reductase enzymes in the bacterial cytosol, initiating the inactivation of the gyrase supercoiling process and altering the ATP-binding site of bacterial gyrase B and DNA. This increases DNA cleavage mediated by topoisomerase IV, leading to deficient bacterial replication and preventing the formation of bacterial biofilms (Er-Rahmani et al., 2024).
The text added about the comparison of the extracts is on page 11, lines 404-413.
Akinduti PA, Osunlola OL, Adebekun FA, Viavonu DT, Elughi GN, Popoola O, Abdulsalami SA. Antibacterial activity of Ocimum sanctum L. essential oil against multidrug resistance bacteria vaginosis. Medicine in Microecology. 2024;22:100115. https://doi.org/10.1016/j.medmic.2024.100115
Er-Rahmani S, Errabiti B, Matencio A, Trotta F, Latrache H, Koraichi SI, Elabed S. Plant-derived bioactive compounds for the inhibition of biofilm formation: a comprehensive review. Environ Sci Pollut Res Int. 2024 May;31(24):34859-34880. Epub 2024 May 15. PMID: 38744766. doi: 10.1007/s11356-024-33532-2.
Comments 13: [Lines 412–417]
The conclusion acknowledges the need for in vivo validation. Do the authors plan to perform cytotoxicity testing on mammalian cell lines or animal model evaluations to confirm the biocompatibility and therapeutic applicability of the pyroligneous extract?
Response 13: This study is part of the ongoing efforts of our research group, which investigates the antimicrobial effects of alternative compounds, such as eucalyptus pyroligneous extract. The data obtained are part of the development of the first author’s doctoral thesis and will serve as a preliminary basis for future studies, including cytotoxicity assays on bovine and caprine epidermal cells and in vivo testing of the extract as an antiseptic during the milking of these animals.
Thanks
Reviewer 3 Report
Comments and Suggestions for Authors
Dear Authors,
Your manuscript „Evaluation of antimicrobial and antibiofilm activity of Eucalyptus urograndis (Clone I144) pyroligneous extract on bovine mastitis isolate of multiple drug resistant Staphylococcus aureus strains“ is well-organized and clearly written, presenting significant findings that could influence the approach to managing Staphylococcus aureus strains responsible for mastitis. The selection and range of tested antibiotics are comprehensive and well-justified. Overall, the study makes a valuable contribution to the chosen topic. I have only two comments regarding this manuscript:
However, for a thorough and well-founded investigation of the use of this extract, it is necessary to include an analysis of its chemical composition. There are no results of GC-MS analyses presented in the manuscript.
The conclusion is very brief; I suggest explaining better the obtained results and providing a more comprehensive summary of the research.
Author Response
Dear Reviewer
We appreciate the review of the article.
Comments 1: Dear Authors,
Your manuscript „Evaluation of antimicrobial and antibiofilm activity of Eucalyptus urograndis (Clone I144) pyroligneous extract on bovine mastitis isolate of multiple drug resistant Staphylococcus aureus strains“ is well-organized and clearly written, presenting significant findings that could influence the approach to managing Staphylococcus aureus strains responsible for mastitis. The selection and range of tested antibiotics are comprehensive and well-justified. Overall, the study makes a valuable contribution to the chosen topic. I have only two comments regarding this manuscript:
However, for a thorough and well-founded investigation of the use of this extract, it is necessary to include an analysis of its chemical composition. There are no results of GC-MS analyses presented in the manuscript .
Response 1: We thank the reviewer for the comments. We agree with this comment and the authors explain the paper did not determine the components of the pyroligneous extract. These components were determined by Gama et al., 2023. The table 2 was inserted in section 2.3, page 5, lines 182-184.
Table 2. Major GC/MS components of the pyroligneous extract of Eucalyptus urograndis I144
|
Major components |
Relative Peak Areas (%) |
|
Furfural |
17.2 |
|
2-Methoxy-phenol |
9.4 |
|
Phenol |
8.5 |
|
5-Methyl-2-Furancarboxaldehyde |
4.4 |
|
2-Methoxy-3-methyl- Phenol |
4.4 |
|
1-(2-Furanyl)-ethanone |
1.5 |
|
2,3-Dimethyl-2-cyclopenten-1-one |
1.4 |
|
2,5-Dihydro-3,5-dimethyl-2-furanone |
1.0 |
|
3-Methyl-2-Cyclopenten-1-one |
2.4 |
|
1,2,3-Trimethoxy-5-methyl-benzene |
1.9 |
|
3-Methyl- Phenol, |
3.3 |
|
1,2,5-Trimethoxybenzene |
3.5 |
|
4-Methyl- Phenol |
2.3 |
|
1,2-Cyclopentanedione-3-methyl |
3.3 |
|
2,6-Dimethoxy- Phenol, |
10.7 |
|
3-ethyl-2-hydroxy-2-cyclopenten-1-one |
1.0 |
|
4-Ethyl-2-methoxy-phenol |
1.8 |
* adapated from table 3 of the paper by Gama et al., 2023
Gama, GSP; Pimenta, AS; Feijó, F; Santos, C; Fernandes, B; Oliveira, M; Souza, E; Monteiro, T; Fasciotti, M; Azevedo, T. Antimicrobial activity and chemical profile of wood vinegar from eucalyptus (E. urophylla × E. grandis – clone I144) and bamboo (Bambusa vulgaris). World J Microbiol Biotechnol 2023, 39, 186. https://doi.org/10.1007/s11274-023-03628-x
Comments 2: The conclusion is very brief; I suggest explaining better the obtained results and providing a more comprehensive summary of the research .
Response 2: We appreciate and agree with the suggestion. The following text has been added to the page 12, lines 451-459 - The results obtained in this study demonstrate that the evaluated extract exhibits significant antimicrobial activity against Staphylococcus aureus strains isolated from bovine mastitis cases, including multidrug-resistant variants and methicillin-resistant S. aureus (MRSA). Furthermore, biofilms were inhibited at different concentrations, ranging from 3.13% (1/6), 6.25% (3/6), to 12.5% (3/6). These findings indicate that the extract represents an effective preventive strategy against mastitis-causing pathogens that are difficult to treat, making it a promising alternative for reducing dependence on synthetic antibiotics. In vivo studies are necessary to confirm these findings and support evidence-based clinical guidelines.
Furthermore, the conclusion was also modified in the abstract, page 1, lines 28-34.
thanks
Round 2
Reviewer 1 Report
Comments and Suggestions for Authors
no comments